# Hepatocrinology

**DOI:** 10.3390/medsci9020039

**Published:** 2021-06-01

**Authors:** Sanjay Kalra, Saptarshi Bhattacharya, Pawan Rawal

**Affiliations:** 1Department of Endocrinology, Bharti Hospital, Karnal 132001, India; 2Department of Endocrinology, Max Hospital, Patparganj, New Delhi 110092, India; saptarshi515@gmail.com; 3Department of Gastroenterology, Artemis Hospital, Gurgaon 122002, India; pawan.rawal@artemishospitals.com

**Keywords:** cirrhosis, diabetes, endocrine, hepatogenous diabetes, liver, NAFLD

## Abstract

Hepatocrinology is defined as a bidirectional, complex relationship between hepatic physiology and endocrine function, hepatic disease and endocrine dysfunction, hepatotropic drugs and endocrine function, and endocrine drugs and hepatic health. The scope of hepatocrinology includes conditions of varied etiology (metabolic, infectious, autoimmune, and invasive) that we term as hepato-endocrine syndromes. This perspective shares the definition, concept, and scope of hepatocrinology and shares insight related to this aspect of medicine. It is hoped that this communication will encourage further attention and research in this critical field.

## 1. Introduction

The liver has long been recognized as the seat of the metabolism. Simultaneously, the endocrine system controls the homeostasis of the body. The subject ‘hepatocrinology’ is the field of medicine that studies the bidirectional relationship between hepatic and endocrine physiology, as well as dysfunction. The discipline of hepatocrinology explores the liver as an endocrine gland by describing its production of hormones and its role in hormonal modulation (by synthesizing transport proteins). The hepato-insular axis is a part of hepatocrine physiology [1]. Endocrine manifestations of liver insufficiency (cirrhosis) and malignancy, and hepatic complications of various endocrine disorders are included. Special attention is paid to hepato-endocrine syndromes in which hepatic and endocrine dysfunction co-occur. The possible hepatotropic effect of endocrine drugs, pleiotropic endocrine consequences of medicines used in the management of liver disease, and potential exaptation of endocrine agents for use in hepatology form part of this science.

## 2. The Liver as an Endocrine Organ

The liver secretes various hormones, which mediate glucose metabolism, blood pressure, growth, and hemorheological homeostasis. These include insulin-like growth factor (IGF)-1, betatrophin, and irisin, all of which mediate insulin sensitivity [2,3]. Angiotensinogen, produced by the liver, is the bedrock of the renin-angiotensin-aldosterone system, which contributes to blood pressure maintenance [4]. Hepcidin and thrombopoietin contribute to the regulation of iron metabolism and platelet production, respectively [5,6]. The hepato-insular axis is a well-researched contributor to glucose metabolism and has been described variously as the entero-insular or adipo-hepato-insular axis [1]. There are several other hormones or their precursors that are synthesized by the liver. Some of the important products are summarized in Table 1 and detailed below.

### 2.1. Insulin-Like Growth Factor and Insulin-Like Growth Factor Binding Proteins

The IGF and IGF-binding proteins (IGFBPs) are primarily synthesized in the liver and constitute a complex system that plays a critical role in cellular proliferation and differentiation [14,15]. Growth hormone (GH), secreted from the somatotrophs in the anterior pituitary, drives the synthesis of IGF-1 in the liver. The IGF-1 is a crucial mediator of development during childhood, and the primary determinant of linear growth. In adults, it continues to exert an anabolic effect, and adult GH deficiency (GHD) portends to a negative cardiovascular (CV) outcome [16]. IGF-2, the other hormone responsible for growth-promoting effects, is widely expressed during fetal development, but synthesized in the liver and epithelial cell lining of the brain surface after birth [17]. The actions of IGF-1 and IGF-2 are modulated both systemically and locally by six different IGFBP subtypes designated IGFBP-1 through 6 [8].

Serum IGF-1 levels are decreased in cirrhosis as the synthetic capacity of the liver is diminished [18,19]. Hepatic IGF-1 production is also lower in those with higher degrees of steatosis, non-alcoholic fatty liver disease (NAFLD) activity score (NAS), and hepatic fibrosis [20,21]. Conversely, NAFLD occurs more commonly in adult GHD. GH and IGF-1 prevent NAFLD by decreasing visceral fat, reducing lipogenesis in the hepatocytes, and improving fibrosis by inactivating stellate cells [22].

The GH-IGF-1 axis is involved in the pathogenesis of several other endocrine and hepatic disorders. Notable among them is the development of hormone-sensitive cancers. There is emerging evidence that cross-talk between sex steroids and IGF-1 modulates the propensity for the development of breast and prostate cancers [23,24]. The various pathophysiological effects of the GH-IGF-1 axis are thus orchestrated through IGF and IGFBP synthesized in the liver.

### 2.2. Angiotensinogen

Hemodynamic homeostasis is governed by hepatic secretory products. A key component among them is angiotensinogen, an alpha-globulin synthesized in multiple tissues [4]. It is abundantly present in the plasma, and the serum levels are determined by hepatic secretion. Renin from the juxtaglomerular cells of the kidney cleaves angiotensinogen to angiotensin I. Renin-mediated cleavage is tightly regulated and considered the rate-limiting step in the production of biologically active angiotensin peptides [25]. Angiotensin I is subsequently converted to angiotensin II by the angiotensin-converting-enzyme (ACE) located predominantly on the endothelial cells of the pulmonary vasculature. Angiotensin II plays a pivotal role in controlling blood pressure and sodium homeostasis through its effect on blood vessels, zona glomerulosa of the adrenal cortex, and the kidney [26]. Additionally, the disequilibrium of the renin-angiotensin-system (RAS) impacts the inflammatory pathways in the lungs and is linked to the development of acute respiratory distress syndrome (ARDS), including the coronavirus disease 2019 (COVID-19) induced lung injury [27,28].

### 2.3. Thrombopoietin

The liver is the source of hematopoietic growth factors and iron transport proteins such as hepcidin. Thrombopoietin, a key hematopoietic cytokine synthesized in the liver, induces megakaryocyte progenitor expansion and differentiation. It additionally assists in the maintenance and expansion of hematopoietic stem cells [6]. Thrombopoietin also determines the lineage of primitive progenitor stem cells and is unique among the hematopoietic cytokines by its effect on both primitive, as well terminally differentiated, cells [29].

### 2.4. Betatrophin and Proprotein Convertase Subtilsin-Kexin Type 9 (PCSK9)

The liver plays a critical role in maintaining lipid balance. Betatrophin, now referred to as angiopoietin-like protein 8 (ANGPTL8), and PCSK9 are critical regulators of lipid metabolism. Though initial reports suggested that betatrophin can stimulate the growth of beta cells of the pancreas, subsequent studies have disproved this [30]. ANGPTL8 modulates the activity of lipoprotein lipase (LPL) through its interaction with ANGPTL3 and stabilizes triglyceride levels [31]. Though the exact mechanism by which this hepatocyte-derived factor regulates metabolism is not clearly understood, it has been linked to obesity, diabetes, hypothyroidism, and polycystic ovary syndrome (PCOS). It has the potential to emerge as a critical therapeutic target in the management of metabolic disorders [32]. PCSK9, synthesized in the liver and several other organs, is a regulatory protein for low-density-lipoprotein (LDL) receptors [33]. The development of PSCK9 inhibitors as lipid-lowering tools is a significant breakthrough in the management of dyslipidemia and atherosclerotic cardiovascular disease [34].

### 2.5. Hormone-Transport Proteins

The liver produces several important proteins that act as carriers for various hormones and thus indirectly modulate critical endocrine functions. Thyroid-binding globulin, transthyretin, and albumin produced in the liver are all involved in the transportation of thyroxine and tri-iodothyronine [10]. Cortisol is mainly bound to corticosteroid-binding globulin, again produced from the liver [35]. Sex hormone-binding globulin not only carries estradiol and testosterone, but can also serve as an early biomarker and a therapeutic target for PCOS [36]. The levels of these proteins are altered in different physiological and pathological states [37,38].

## 3. Endocrine Manifestations of Hepatic Disease

The liver modulates the functioning of the endocrine system directly or indirectly in multiple ways. Liver dysfunction is thus predictably associated with various endocrine disorders. The significant anomalies have been detailed below and depicted in Figure 1.

### 3.1. Insulin Resistance and Diabetes

Diabetes is a leading cause of the development of NAFLD and cirrhosis. On the other hand, cirrhosis causes insulin resistance and increases the probability of developing diabetes, with a reported prevalence ranging from 30–70% in different studies [39,40,41]. The hyperglycemia arising from liver dysfunction is referred to as hepatogenous diabetes and is pathophysiologically distinct from type 2 diabetes mellitus (T2DM) [42]. The fasting plasma glucose (FPG) and glycated hemoglobin (HbA1c) are often normal in hepatogenous diabetes, and an abnormal oral glucose tolerance test (OGTT) is usually required to establish the diagnosis [43]. The mechanism of the development of diabetes. in cirrhosis is complex and only partially understood. Insulin resistance from altered secretion of adipokines, inflammatory cytokines, incretins, and free fatty acids play a significant contributory role [43,44].

Additionally, hypoxia-inducible factors and advanced glycosylation end-products (AGEs) can result in impaired insulin secretion [43,45]. Hepatitis C virus itself decreases insulin sensitivity by altering insulin signaling and increasing endoplasmic reticulum stress [46,47,48]. Hepatic and systemic insulin resistance often precedes the onset of cirrhosis and is present in individuals with NAFLD [49,50].

### 3.2. Hypoglycemia

Hypoglycemia is commonly encountered in patients with advanced cirrhosis, especially if a concurrent infection is present [51,52]. Hypoglycemia occurs in up to 40% of cases of acute liver failure and is associated with increased mortality [53,54]. The mechanism behind hypoglycemia is depletion of glycogen stores, decreased gluconeogenesis, and impaired insulin clearance by the liver [55,56]. Non-islet cell tumor hypoglycemia (NICTH) is a rare paraneoplastic manifestation of hepatocellular carcinoma (HCC) [57]. Low serum insulin, C-peptide, and beta-hydroxybutyrate in combination with high IGF-2 characterize NICTH [58].

### 3.3. Gonadal Dysfunction

Hypogonadism and gynecomastia are well-recognized manifestations of cirrhosis of the liver. The possible mechanisms include decreased production of sex hormone-binding globulin, decreased hepatic clearance of estrogen, primary testicular defect, hypothalamic-pituitary dysfunction, and direct toxic effects of alcohol on gonads [59,60]. Women with cirrhosis can manifest menstrual irregularities such as oligomenorrhea or amenorrhea, primarily resulting from hypothalamic-pituitary dysfunction [61]. Undernutrition and elevated serum prolactin can also produce irregularities in the menstrual cycle [62].

Cirrhosis in men can manifest with features of hypogonadism such as loss of secondary sexual characters and decreased libido [63]. Gynecomastia is reported in up to 44% of men with cirrhosis, and ascribed to the elevated estrogen:testosterone ratio [64,65]. Testosterone levels are low in patients with cirrhosis, and progressively decrease while the severity of the liver disease increases [66]. Low testosterone is responsible for body hair loss, sarcopenia, osteoporosis, anemia, and fatigue, and is a marker of increased mortality in cirrhosis [67,68]. Primary hypogonadism, indicated by an elevation in serum levels of luteinizing hormone (LH), can occur in alcohol-induced cirrhosis. It can be attributed to the direct toxic effect of alcohol on the testis [69,70]. Hypogonadotropic hypogonadism that partially reverses after liver transplantation is described in most other forms of cirrhosis [65]. Low testosterone levels stimulate the synthesis of sex-hormone-binding globulin (SHBG) in cirrhosis. SHBG levels are elevated in liver disease, except in advanced stages where the synthetic capacity of the liver is diminished [66].

### 3.4. Skeletal Manifestations

Alteration in bone metabolism generally occurs in cirrhosis of the liver. Hepatic osteodystrophy refers to the skeletal manifestations of cirrhosis and encompasses osteoporosis and, in rare cases, osteomalacia and rickets [71]. The metabolic bone disease in cirrhosis is multifactorial and results from nutritional factors, proinflammatory state, synthetic defects, and hypogonadism [72]. Primary biliary cirrhosis (PBC) has been mainly linked to a low bone-turnover state resulting from decreased production of growth factors such as IGF-1, elevated levels of lithocholic acid (known to prevent osteoblast formation), and vitamin K deficiency [73]. Osteoporosis and fragility fractures are recognized but under-diagnosed complications of cirrhosis, and can be prevented by early diagnosis and treatment [74].

### 3.5. Thyroid Disorders

The autoimmune disorders often tend to coexist, and thyroid dysfunction and high prevalence of thyroid autoantibodies have been observed in autoimmune hepatitis, PBC, and primary sclerosing cholangitis [75]. Hepatitis C infection is also associated with the development of thyroid disorders [76]. A meta-analysis of five studies after adjusting for heterogeneity suggested that hepatitis C infection increased the chance of the development of thyroid cancer [77]. The serum concentration of thyroid-binding globulin (TBG) is elevated in HCC, and normalizes after resection of the tumor [78,79].

### 3.6. Adrenal Insufficiency

Adrenal insufficiency is reported in patients with cirrhosis during septic shock and decompensated liver disease [80]. The term hepato-adrenal syndrome has been used to define relative adrenal insufficiency occurring in patients with cirrhosis. Though the exact mechanism is not clearly understood, diminished hepatic synthesis of cholesterol resulting in the deficiency of substrate for steroid synthesis in the adrenal cortex is a proposed hypothesis [81].

### 3.7. Growth Disorders in Children

Children with cirrhosis commonly exhibit restricted linear growth [82]. Even though GH levels are high in cirrhosis, decreased IGF-1 and IGFBP3 synthesis by the liver induce growth hormone resistance. Thus, administration of exogenous growth hormone has minimal benefit in children with cirrhosis and short stature [83]. Liver transplantation partially restores linear growth rate, but delayed puberty and reduced final adult height are still common [84].

## 4. Hepatic Manifestations of Endocrine Disease

Endocrine and metabolic diseases are a common cause of hepatic dysfunction. The common endocrine causes of liver dysfunction have been depicted in Table 2. NAFLD resulting from metabolic disorders such as diabetes, obesity, and dyslipidemia has emerged as one of the leading causes of chronic liver disease over the past two decades. Several other hormonal disturbances affect the functioning of the liver directly or indirectly.

### 4.1. Non-Alcoholic Fatty Liver Disease

NAFLD has a bidirectional and complex relationship with metabolic syndrome and insulin resistance. NAFLD refers to a group of disorders characterized by fat accumulation in the liver in the absence of other secondary causes. The spectrum of NAFLD encompasses steatosis or steatohepatitis with associated fibrosis, and can progress to cirrhosis. The risk for HCC is also elevated in patients with NAFLD [85]. Insulin resistance, a key component of metabolic syndrome, plays an essential role in the pathogenesis of NAFLD [86]. Obesity, T2DM, and dyslipidemia are strongly associated with the development of NAFLD, though the exact pathophysiologic link is a subject of research [87].

Several factors such as genetic and epigenetic factors, nutrition, adipose tissue dysfunction, gut microbiota, inflammation, oxidative stress, adipocytokines, and hepatic iron have been implicated, however the influence of insulin resistance in the pathogenesis of NAFLD remains central [88]. Uninhibited adipose tissue lipolysis resulting from systemic insulin resistance, coupled with increased lipogenesis leads to increased delivery and deposition of free fatty acids in the liver [89]. The toxicity of accumulated lipids in hepatic cells triggers further inflammation and damage. Free fatty acids stimulate endoplasmic reticulum stress and mitochondrial pathways of apoptosis. Lipoapoptosis induces hepatic fibrosis and further progression to cirrhosis [90].

In recent years, NAFLD and non-alcoholic steatohepatitis has emerged as an important risk factor for development of HCC even in the absence of cirrhosis [91]. The carcinogenesis results from alteration in complex signaling pathways mediated by genetic, immunologic, metabolic, and endocrine interactions [92]. Insulin resistance and hyperinsulinemia associated with NAFLD augment IGF-1 synthesis in the liver [93]. Stimulation of insulin receptor and IGF-1 receptor initiates insulin receptor substrate-1 pathway activation and subsequent downstream induction of PI3K and MAPK pathways [94]. The activation of these pathways induce cell proliferation, prevent apoptosis, and act as the link between insulin resistance and carcinogenesis of HCC [95].

NAFLD is the leading cause of chronic liver disease in many parts of the world and metabolic syndrome, diabetes, and obesity remain its primary drivers [96]. The strong connection between insulin resistance and NAFLD, NASH, and HCC reinforces the importance of the intricate relationship between endocrine pathways and liver.

### 4.2. Secondary NAFLD from Other Endocrine Disorders

Steatosis or steatohepatitis has been observed in multiple other endocrine anomalies such as hypothyroidism, Graves’ disease and other causes of thyrotoxicosis, PCOS, Cushing’s syndrome, acromegaly, and pheochromocytoma [97]. Hypothyroidism is a risk factor for NAFLD. A recent meta-analysis of 26 studies demonstrated that thyroid stimulating hormone (TSH) levels can correlate with development and progression of NAFLD [98]. However, other reports did not establish the link [99]. Such an association is mechanistically plausible given the effect of thyroid hormone on fat deposition in the liver and other body parts [100]. The prevalence of NAFLD is reported to be only 20% in Cushing’s syndrome, in spite of the presence of several features of metabolic syndrome such as central obesity and insulin resistance [101]. The low prevalence of NAFLD could result from the immunosuppressive effect of cortisol, especially the low grade chronic inflammation mediated by interleukin-6 [102]. PCOS is also associated with NASH, and the two conditions share common genetic and metabolic factors [103]. GH deficiency also increases the risk of NAFLD as already discussed in the previous section.

### 4.3. Other Hepatic Manifestations of Endocrine Disorders

The liver can be the site of metastases for many endocrine cancers such as adrenal carcinoma, pancreatic carcinoma, and testicular and ovarian tumors [104]. The unique constellation of clinical symptoms observed in carcinoid syndrome usually occurs after extensive hepatic metastases from gastrointestinal carcinoids. The liver otherwise metabolizes the bioactive products secreted into the portal circulation by the tumors [105]. Cholestasis can be a hepatic manifestation of thyroid disorders [106]. Neonatal cholestasis can be an indicator of the presence of congenital combined pituitary hormone deficiency or congenital hypothyroidism [107,108].

## 5. Sexual Dimorphism in Liver Disorders

Many liver diseases show differential gender distribution. NAFLD is more common in men during the reproductive age group, but is more frequent in women after menopause, indicating a possible protective role of estrogen [112]. HCC occurs more commonly in men, while the risk of autoimmune liver diseases such as primary biliary cirrhosis and autoimmune hepatitis is more common in women [113]. Women also show higher vulnerability to alcohol-related liver diseases [114]. Apart from sex hormones, differences in xenobiotics, immune function, genetic alterations, and receptor expression are presumed to drive the dichotomy [115].

## 6. Liver Function Biochemical Markers as Predictors of Endocrine Dysfunction

In several studies, liver enzymes have correlated with the development of incident diabetes [116]. γ-glutamyltransferase (GGT) has been proposed as a marker of oxidative stress and is associated with the future risk of diabetes. GGT levels have also been considered an indicator of hepatic fat deposition, which is related to insulin resistance [117]. In several reports, GGT and alanine aminotransferase in early pregnancy predicted the future occurrence of gestational diabetes mellitus [118,119]. Table 3 summarizes the liver enzymes which have been linked to the future development of metabolic disorders.

## 7. Hepato-Endocrine Syndromes

We have used the term “hepato-endocrine syndromes” to describe disorders with a common etiology that manifest as combined hepatic and endocrine dysfunction. The various hepato-endocrine syndromes are enumerated in Table 4. Disorders of iron and copper metabolism such as hemochromatosis and Wilson’s disease are notable examples of this syndrome [121,122]. Polyglandular autoimmune syndromes type 1 and type 2 can develop autoimmune hepatitis and primary biliary cirrhosis, respectively, as their hepatic manifestations [123]. Hepatitis C virus infection can be associated with thyroiditis and hypothyroidism [124].

## 8. Hepatic Effect of Endocrine Drugs

The endocrine drugs can have harmful as well as beneficial effects on the liver. Both anabolic steroids and estrogens can cause cholestasis, hepatic adenoma, focal nodular hyperplasia, and other hepatic disorders [128,129]. Acute liver failure has been reported with diverse agents such as propylthiouracil (used for hyperthyroidism) and high doses of methylprednisolone [130,131]. Orlistat, a commonly used therapy for weight loss, has also been described to cause subacute and acute liver failure [132].

On the other hand, the anti-diabetic agents such as pioglitazone and possibly sodium-glucose cotransporter-2 (SGLT2) inhibitors and glucagon-such as peptide-1 receptor agonist (GLP1RA) might possibly have a beneficial effect on NAFLD [133]. Glucocorticoid is indicated for the treatment of autoimmune hepatitis [134]. Somatostatin and vasopressin analogs decrease portal blood flow and help control esophageal variceal bleeding [135].

## 9. Endocrine Effects of Drugs Used in Hepatology

Spironolactone, commonly used for the management of ascites in patients with cirrhosis, is an anti-androgen which has beneficial effects in PCOS in women, but causes painful gynecomastia in males [136,137]. Interferon-alpha used for management of hepatitis C infection can result in thyroid dysfunction [138]. Beta-blockers have often been associated with erectile dysfunction [139]. Table 5 depicts the common drug interactions in hepatocrinology.

## 10. Conclusions

The spectrum of hepatocrinology envelops diverse interactions between hepatic and endocrine systems in health and disease. We have coined this portmanteau term to increase awareness among clinicians about the complex, multifaceted relationships between these two disciplines. Both diabetes and NAFLD are emerging epidemics, and early recognition of the interconnection between these commonly prevalent disorders might assist in preventing advanced complications such as cirrhosis and HCC. Chronic liver disease results in multiple endocrine dysfunctions in all stages of life. Children with cirrhosis have stunted linear growth; in reproductive age groups hypogonadism remains a concern; and the elderly are affected by osteoporosis. Many of these relations remain unappreciated, and the complications undiagnosed in clinical practice. We hope that the study of hepatocrine interplay under a well-structured rubric will make clinicians aware of these often-missed interactions and improve patient outcome.

## Figures and Tables

**Figure 1 medsci-09-00039-f001:**
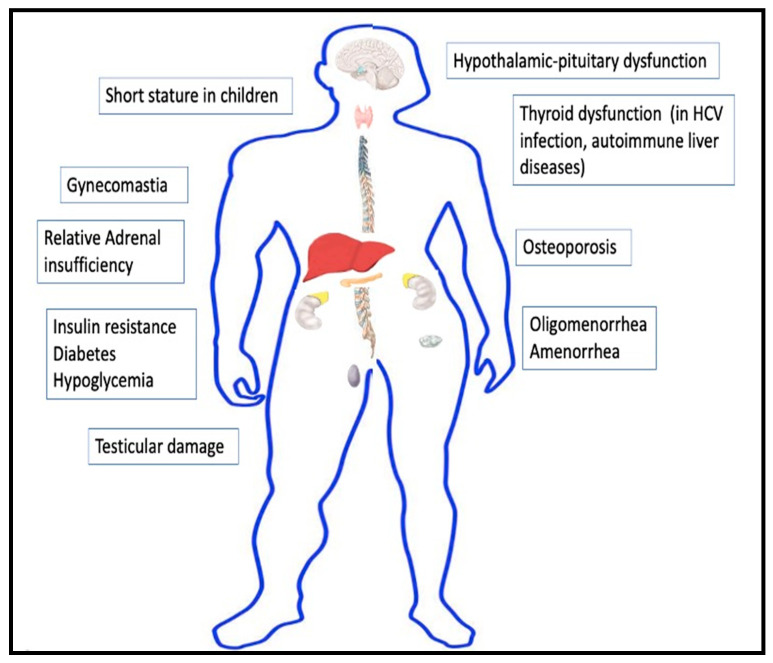
Endocrine manifestations of cirrhosis.

**Table 1 medsci-09-00039-t001:** The liver as an endocrine organ.

Action	Hormones	Reference
Hormone synthesis	IGF-1	Bach [2]
	Angiotensinogen	Matsuaska [4]
	Thrombopoietin	Hitchcock [6]
	Hepcidin	Ruchala [5]
	Betatrophin	Raghow [3]
	Proprotein convertase subtilsin-kexin type 9	Yadav [7]
Hormone action modulation	IGF binding protein 1 to 6	Allard [8]
	Sex hormone-binding globulin	Selby [9]
	Thyroid hormone-binding globulin	Schussler [10]
	Transthyretin	Palha [11]
	Corticosteroid binding globulin	Breuner [12]
	Vitamin D binding protein	Bouillon [13]

IGF—insulin-like growth factor.

**Table 2 medsci-09-00039-t002:** Hepatic manifestations of endocrine disorders.

Hepatic Manifestation	Endocrine Disorders	References
Non-alcoholic fatty liver disease	Insulin resistance, diabetes, obesity, and dyslipidemia	Watt [86]
Hepatic steatosis or steato-hepatitis	Cushing’s syndrome, acromegaly, Graves’ disease and other causes of thyrotoxicosis, polycystic ovary syndrome, male hypogonadism, and pheochromocytoma	Lonardo [97]
Hepatic metastasis	Adrenal cancer, pancreatic cancer, ovarian and testicular neoplasm, and malignant pheochromocytoma	Ridder [104]
Neonatal cholestasis	Congenital combined pituitary hormone deficiency, congenital hypothyroidism, and HNF1B-MODY (previously MODY-5)	Chan [107], Korkmaz [108]
Acute hepatic congestion (with jaundice)	Myxedema coma	Villalba [109]
Cholestasis	Thyrotoxicosis	Abebe [110]
Congestive hepatomegaly	Thyrotoxic heart failure	Piantanida [106]
Mauriac syndrome	Poorly controlled diabetes mellitus	Subedi [111]

HNF—hepatocyte nuclear factor, MODY—maturity-onset diabetes of young.

**Table 3 medsci-09-00039-t003:** Liver function biochemical markers as predictors of endocrine dysfunction.

Abnormality in Liver Function	Significance	References
Raised GGT	Probable role in the prediction of future risk of diabetes	Kaneko [116]
Elevated ALT	Probable role in the prediction of future risk of diabetes	Kaneko [116]
Elevated ALT and GGT in early pregnancy	Correlates with development of gestational diabetes mellitus	Lee [118], Zhao [119]
Elevated liver enzymes	Possible marker of insulin resistance and metabolic syndrome	Marchesini [120]

GGT—γ-glutamyltransferase, ALT—alanine aminotransferase.

**Table 4 medsci-09-00039-t004:** Hepato-endocrine syndromes.

**Disease**	**Hepatic Manifestation**	**Endocrine Dysfunctions**
**Metabolic disorders**		
Hemochromatosis [121]	Hepatic fibrosis, cirrhosis, and hepatocellular carcinoma	Diabetes, hypopituitarism, secondary hypogonadism, and secondary hypothyroidism
Wilson’s disease [122]	Transaminitis, steatosis, acute hepatitis and acute liver failure (with an associated Coombs-negative hemolytic anemia), chronic hepatitis, and cirrhosis	Fanconi syndrome, distal renal tubular acidosis, nephrolithiasis, gigantism, hypoparathyroidism, pancreatitis, impotence, infertility, and repeated spontaneous abortions
Glycogen storage disorders: Glycogen storage disease I (von Gierke disease)—90% of cases [125]	Glucose-6-phosphatase deficiency in liver and muscle, hepatomegaly, and hepatic adenomas	Hypoglycemia, lactic acidosis, hypertriglyceridemia, and hyperuricemia; short stature, and delayed puberty
**Autoimmune disorders**		
Polyglandular autoimmune syndrome 1 [123]	Autoimmune hepatitis	Hypoparathyroidism and autoimmune adrenal insufficiency (along with chronic mucocutaneous candidiasis)
Polyglandular autoimmune syndrome 2 [123]	Primary biliary cirrhosis	Addison’s disease plus either an autoimmune thyroid disease or type 1 diabetes mellitus associated with hypogonadism, and other endocrinopathies
**Infections**		
Hepatitis C infection [124]	Chronic hepatitis C, cirrhosis, and hepatocellular carcinoma	Thyroid autoimmunity, hypothyroidism, and higher prevalence of thyroid cancer
Hepatitis B infection [126]	Chronic hepatitis B, cirrhosis, and hepatocellular carcinoma	Increased risk of diabetes mellitus
**Malignancy**		
Paraneoplastic endocrine syndromes [127]	Hepatocellular carcinoma	Hypoglycemia, hypercholesterolemia, and hypercalcemia

**Table 5 medsci-09-00039-t005:** Pharmacological interactions in hepatocrinology.

**Hepatic Effects of Endocrine Drugs**	
**Drugs**	**Adverse Effects**
Anabolic androgenic steroid [128]	Hepatic adenoma, hepatocellular carcinoma, cholestasis, and peliosis hepatis.
Estrogen/oral contraceptive pills [129]	Intrahepatic canalicular cholestasis, hepatic adenomas, focal nodular hyperplasia, hemangioma or hamartoma, peliosis hepatis, and Budd Chiari syndrome
Tamoxifen [140]	NAFLD
Propylthiouracil, methimazole, carbimazole [130]	Hepatitis, cholestasis, and acute liver failure
Corticosteroids [131]	Hepatic enlargement, steatosis, glycogenosis. NAFLD, exacerbate chronic viral hepatitis, and high doses of intravenous methylprednisolone—acute liver failure (sometimes fatal)
Vasopressin receptor antagonist [141]	Transaminitis and acute liver failure
Orlistat [132]	Cholelithiasis, cholestatic hepatitis, and acute and subacute liver failure
**Drugs**	**Beneficial effects**
Pioglitazone [133]	Beneficial effect on NAFLD
GLP-1RA [133]	Possible beneficial effect on NAFLD
SGLT-2 inhibitors [133]	Possible beneficial effect on NAFLD
Saroglitazar [133]	Possible beneficial effect on NAFLD
Corticosteroids [134]	Treatment of autoimmune hepatitis and prevention of rejection of liver transplant
Somatostatin analogs (octreotide and others) [135]	Treatment of variceal bleeding (decreases portal blood flow)
Vasopressin analogs (terlipressin) [135]	Treatment of variceal bleeding (decreases portal blood flow)
**Endocrine Effects of Drugs Used in Hepatology**	
**Drugs**	**Adverse effects**
Spironolactone [137]	Gynaecomastia, and hypogonadism in men
Beta-blockers [139]	Erectile dysfunction
Interferon-alpha [138]	Hypothyroidism, autoimmune (Hashimoto’s) thyroiditis, destructive thyroiditis, and Graves’ disease
**Drugs**	**Beneficial effects**
Ursodeoxycholic acid [142]	Possible beneficial effect in metabolic syndrome
Spironolactone [136]	Treatment of PCOS

NAFLD—non-alcoholic fatty liver disease, PCOS—polycystic ovary syndrome, GLP-1RA glucagon-like peptide receptor agonist, SGLT-2—sodium glucose cotransporter-2, GGT—γ-glutamyltransferase, and ALT—alanine aminotransferase.

## Data Availability

Not applicable.

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
