# Peer review of "Hepatocrinology"

_medsci, 2021, doi:10.3390/medsci9020039_

Round 1
Reviewer 1 Report
The authors are proposing a new term, called “hepotocrinology”. They define the term and discuss each aspect to be addressed by “hepatocrinology” in a succinct way. The liver is vulnerable to a wide variety of metabolic changes from different sources. The study of the liver as an endocrine gland could facilitate the understanding of liver dysfunctions, which usually occur simultaneously with endocrine changes. However, some points in the manuscript deserve attention:
- All tables: references must be indicated in the table, for example in a specific column.
- Table 2: This table does not make sense. It should be replaced by a figure.
- Conclusion section: The new term draws attention to the endocrine and hepatic issue. However, it was not clear how this new term could result in improvements for the training of new clinicians, and also for the improvement of the patient. Please clarify this point.
Reviewer 2 Report
the subject is quite interesting and leaves the way open for further reviews and also clinical studies and animal models.
The authors must do the following changes for a better manuscript:
Table 2: is not complete, and has extra phrases of “reference in text”
Item 4. Must say “Table 3” instead a “Table”.
Item 5. Please change the word “Indices” for “biochemical markers”.
Item 6. Line 2 has an extra space (next to etiology).
Item 7. Put the respective acronymous for GLP-1 and SGLT.
Round 2
Reviewer 1 Report
The work shares the definition, concept and scope of hepatocrinology, and shares insight related to this aspect of medicine. But, the impact of this new perspective is not yet clear.
Figure 1 needs to be improved.